# Effects of Whey and Pea Protein Supplementation on Post-Eccentric Exercise Muscle Damage: A Randomized Trial

**DOI:** 10.3390/nu12082382

**Published:** 2020-08-09

**Authors:** David C. Nieman, Kevin A. Zwetsloot, Andrew J. Simonson, Andrew T. Hoyle, Xintang Wang, Heather K. Nelson, Catherine Lefranc-Millot, Laetitia Guérin-Deremaux

**Affiliations:** 1Human Performance Laboratory, Department of Biology, Appalachian State University, North Carolina Research Campus, Kannapolis, NC 28608, USA; simonsonaj@appstate.edu (A.J.S.); hoylea@appstate.edu (A.T.H.); 2Department of Health and Exercise Science, Appalachian State University, Boone, NC 28608, USA; zwetslootka@appstate.edu; 3China Academy of Sport and Health Sciences, Beijing Sport University, Beijing 100084, China; wangxintang@bsu.edu.cn; 4Nutrition and Health Research & Development, Roquette, Geneva, IL 60134, USA; heather.nelson@roquette.com; 5Nutrition and Health Research & Development, Roquette, 62136 Lestrem, France; catherine.lefranc-millot@roquette.com (C.L.-M.); laetitia.guerin-deremaux@roquette.com (L.G.-D.)

**Keywords:** protein, exercise, muscle damage, creatine kinase, myoglobin, inflammation

## Abstract

This randomized trial compared pea protein, whey protein, and water-only supplementation on muscle damage, inflammation, delayed onset of muscle soreness (DOMS), and physical fitness test performance during a 5-day period after a 90-min eccentric exercise bout in non-athletic non-obese males (*n* = 92, ages 18–55 years). The two protein sources (0.9 g protein/kg divided into three doses/day) were administered under double blind procedures. The eccentric exercise protocol induced significant muscle damage and soreness, and reduced bench press and 30-s Wingate performance. Whey protein supplementation significantly attenuated post-exercise blood levels for biomarkers of muscle damage compared to water-only, with large effect sizes for creatine kinase and myoglobin during the fourth and fifth days of recovery (Cohen’s d > 0.80); pea protein versus water supplementation had an intermediate non-significant effect (Cohen’s d < 0.50); and no significant differences between whey and pea protein were found. Whey and pea protein compared to water supplementation had no significant effects on post-exercise DOMS and the fitness tests. In conclusion, high intake of whey protein for 5 days after intensive eccentric exercise mitigated the efflux of muscle damage biomarkers, with the intake of pea protein having an intermediate effect.

## 1. Introduction

Skeletal muscle comprises ~40–45% of the total mass and ~60% of the total body protein in humans, and accounts for ~30–45% of whole-body protein metabolism [1,2]. Most adults need no more than 0.8 to 0.9 g protein/kg per day to satisfy protein needs [1,2,3]. The protein needs of athletes are higher than those of sedentary persons, ranging between 1.2 and 2.0 g protein/kg per day, with an intake of 2.0 g/kg per day linked to maximal whole-body anabolism in resistance-trained men after exercise [3]. Most investigators report an additive anabolic effect of protein supplementation (0.25 g high-quality protein/kg or 20–40 g) after an acute resistance exercise session, but this response varies markedly between individuals [4,5,6,7,8,9,10,11,12,13]. 

The optimal protein dosing regimen for increased rates of myofibrillar protein synthesis (MPS) and gains in fat-free mass (FFM) and strength in response to resistance training is still being debated [13,14,15,16,17,18,19,20,21,22,23,24,25,26]. The leucine content of a protein supplement appears to be a strong determinant, and a leucine threshold of 700–3000 mg per protein supplement dose has been recommended [12,13]. The frequency and timing of protein supplement ingestion also play a role, and data support a strategy of spreading intake throughout the day, especially before and immediately after exercise, and before bedtime [13,25,26]. Isotopic tracer studies indicate that leucine-rich whey protein is rapidly digested, promoting a protein anabolic response when ingestion takes place post-exercise and before sleep [13,14,15,16,17,18,26]. Studies using L-[1-^13^C]-phenylalanine-labeled protein indicate that at least half of dietary protein is digested, absorbed, and made available to peripheral tissues, such as skeletal muscle, and that whey protein is more rapidly digested and absorbed compared with micellar casein [19].

Intense eccentric resistance exercise produces high mechanical forces, leading to muscle damage, soreness, inflammation, and a loss of muscle function [27]. Protein supplements are frequently consumed by athletes in the belief that they will enhance recovery from intense exercise by stimulating muscle protein synthesis and remodeling, and thereby alleviate muscle damage and soreness [27,28]. Published studies and reviews, however, do not provide a clear consensus that protein supplementation before, during, and following eccentric resistance exercise reduces muscle damage and accelerates performance recovery [27,28,29,30,31,32]. Challenges in this research area include the 1) appropriate research design (type of participants, nature of the exercise challenge bout, duration of monitoring); 2) highly variable responses between study participants in the release of surrogate markers of exercise-induced muscle injury, including creatine kinase and myoglobin; and 3) the optimum protein dosing regimen and type of protein supplement. The repeated bout effect precludes the use of randomized crossover trials when evaluating the countermeasure effect of protein supplementation on post-exercise muscle damage biomarkers [27].

Intake of plant protein has increased during the past two decades in part due to environmental advantages and the linkage to improved health and decreased all-cause mortality [33,34]. Pea protein isolate from the yellow pea (*Pisum sativum*) contains 85% protein, 9% fat, 0% carbohydrate, and 4% ash. Table 1 compares the amino acid composition for 100 g of the pea protein and whey protein supplements used in this trial [35]. One study showed that 50 g/day doses of pea or whey protein during a 12-week resistance training period resulted in similar increases in muscle thickness relative to the placebo [35]. Few studies have evaluated the effect of plant protein ingestion on exercise-induced muscle damage. Supplementation with oat protein (25 g/day, 18 days) in one study was linked to a decrease in delayed onset of muscle soreness (DOMS), serum creatine kinase and myoglobin, and serum C-reactive protein (CRP) after an intense downhill treadmill run [36].

This randomized trial compared pea protein, whey protein, and water-only supplementation on muscle damage, DOMS, inflammation (CRP), and exercise performance during a 5-day period after a 90-min whole-body eccentric exercise bout in non-athletic non-obese males. We utilized a randomized parallel group design, and emphasized high amounts of supplemental whey and pea protein split into three doses per day for several days post-exercise. The three daily acute pea and whey protein doses (0.3 g/kg) differed in leucine content but both provided amounts within the recommended leucine intake range of 700–3000 mg [13]. Thus, we hypothesized that large daily doses (0.9 g/kg) of pea and whey proteins compared to water spread throughout each day of the 5-day period would counter exercise-induced muscle damage, DOMS, and performance decrements. 

## 2. Materials and Methods 

### 2.1. Study Participants

Study participants were invited to take part in this study if they met the inclusion criteria: Male 18 to 55 years of age, not engaged in regular resistance training (less than 3 sessions per week), body mass index (BMI) under 30 (non-obese), healthy and at low-risk status for cardiovascular disease, and willingness to avoid the use of protein and large-dose vitamin/mineral supplements (above 100% of the recommended dietary allowances), herbs, and all medications (in particular, non-steroidal anti-inflammatory drugs or NSAIDs, such as ibuprofen and aspirin) during the week of the project. Participants voluntarily signed the informed consent, and procedures were approved by the university Institutional Review Board (18-0165). Trial Registration: ClinicalTrials.gov, U.S. National Institutes of Health, identifier: NCT03448328. 

The Consolidated Standards of Reporting Trials (CONSORT) flow diagram is shown in Figure 1. After *n* = 136 participants were assessed for eligibility, *n* = 109 were randomized by A.J.S. to one of three groups (whey protein, pea protein, water), and *n* = 92 received and completed the allocated intervention.

### 2.2. Study Design and Protein Supplementation Protocol

Participants reported to the Human Performance Laboratory (HPL) at the North Carolina Research Campus, Kannapolis, NC, prior to the start of the study for an orientation to the study procedures and body composition testing. Body composition was measured with the Bod Pod body composition analyzer (Life Measurement, Concord, CA, USA). The participants were randomized to one of three parallel groups using a 1:1 allocation (randomization.com): Pea protein, whey protein, or water only. Data for this study were collected during 2018 and 2019. Study participants continued their normal physical activity and food intake schedules during the 5-day study. Physical activity and food intake were not measured before or during the study.

Participants returned to the HPL in an overnight fasted state for day 1 (Mondays) sample collection, physical fitness testing, and the eccentric exercise bout. Participant ratings of the delayed onset of muscle soreness (DOMS) scale (1–10) were recorded prior to blood sampling [37]. Protein supplements (0.3 g protein/kg body mass) with 237 mL of water or water alone (237 mL) were ingested. Pea protein isolate (85.2% protein) (NUTRALYS^®^S85 Plus, Roquette, Lestrem, France) and whey protein isolate (97.4% protein) (biPro, Eden Prairie, MN, USA) were administered under double blind procedures in shaker bottles. Gram amounts for the protein isolates were adjusted for the differences in protein percentages to ensure that exactly 0.3 g of pea or whey protein/kg were ingested. The average dietary protein intake using national data for the age group selected for this study (ages 18 to 55 years) is 103 ± 4.3 g or approximately 1.25 g/kg per day for non-obese males [38]. Thus, dietary intake for protein combined with the supplement was calculated at the high but safe level of 2.15 g/kg per day, well below the 3.5 g/kg tolerable upper level [39]. 

Physical fitness testing was initiated within 5 min after the supplements were ingested. Participants were tested for bench press lifting performance, vertical jump performance, leg-back strength, and anaerobic power with the 30-s Wingate test. In the bench press to exhaustion, participants laid down supine on the bench, and bench pressed a weighted bar equal to 75% of their body weight as many times as possible at a rate of 30 lifts per minute. In the vertical jump test, participants jumped as high as possible with one hand, and tapped the measuring device (Vertec vertical jump apparatus, Questtek Corp, Northridge, CA, USA). This test was repeated three times, with the best score recorded as the difference between the jump and standing reach heights. Leg/lower back strength was assessed with a dynamometer (Lafayette Instruments, Lafayette, IN, USA). With the legs slightly bent at the knee, participants grasped a bar attached via a chain to the dynamometer with straight arms, and then lifted with maximal effort for several seconds. This test was repeated three times, with the best score recorded. The Lode cycle ergometer (Lode B.V., Groningen, Netherlands) was used for the 30-s Wingate cycling test. The workload was adjusted to the body mass of the subject (0.7 Newton meters per kilogram), and participants cycled at maximal speed for 30 s. The peak and total wattage power output were recorded and adjusted to body mass.

Study participants next engaged in a 90-min eccentric exercise bout (Table 2) that consisted of 16 different exercises, most with 2 to 3 sets and 30–60 s of rest between sets [40]. 

Immediately following the 90-min eccentric exercise bout, study participants provided a muscle soreness rating and a blood sample, ingested another protein dose (0.3 g/kg) (or water), and then repeated the four physical fitness tests. Study participants refrained from any food or beverage intake (except for water) for one hour after taking the second supplement dose. Participants were supplied with protein supplements or just water in a shaker bottle (0.3 g/kg in one cup of water) and told to consume these just before going to bed that Monday evening.

Participants returned at 7:00 a.m in an overnight fasted state four days in a row (Tuesday through to Friday) after the eccentric exercise bout (Monday), and provided each morning a DOMS rating and blood sample followed by ingestion of protein (0.3 g/kg) with water or water only. Following ingestion, participants repeated the four physical fitness tests. Participants ingested another protein dose (0.3 g/kg in one cup of water) (or water) immediately after the physical fitness tests, and then again just before going to bed each evening (Tuesday, Wednesday, Thursday). The total supplemental protein ingested during the week (13 doses) was 3.9 g/kg. 

### 2.3. Blood Sample Analysis

The primary outcome measures were serum muscle damage biomarkers, and the secondary outcome measure was C-reactive protein (CRP) (inflammation). Blood samples were collected in serum separator tubes, centrifuged, and analyzed (same day analysis) for serum myoglobin, creatine kinase, lactate dehydrogenase (LDH), CRP, and comprehensive diagnostic chemistries by a commercial laboratory (LabCorp, Burlington, NC, USA).

### 2.4. Statistical Analysis

The study participant number (*n* = 30 or 31 per group) provided > 80% power to detect a group difference for creatine kinase and myoglobin with effect sizes > 0.70 at alpha 0.05 using two-sample *t*-tests. The data are expressed as mean±SE and were analyzed using the generalized linear model (GLM), repeated measures ANOVA module in SPSS (IBM SPSS Statistics, Version 24.0, IBM Corp, Armonk, NY, USA). The statistical model utilized the between-subjects approach: 3 (groups) × 6 (time points) repeated measures ANOVA. This paper provides the time effect (collective effect of the eccentric exercise bout), the treatment effect (whether the treatment groups differed, main effect), and the interaction effect (whether the data pattern over time differed between groups). If the interaction effect was significant (*p* < 0.05), then post-hoc analyses were conducted using Student’s *t*-tests comparing time point contrasts (A to B, A to C, A to D, A to E, A to F) between groups. An alpha level of *p* < 0.01 was used after Bonferonni correction for 5 multiple tests. Cohen’s d was calculated as the difference between the means divided by the pooled standard deviation, with effect sizes regarded as small, medium, and large using cutoff magnitudes of 0.20, 0.50, and 0.80, respectively. 

## 3. Results

The characteristics for the participants completing all aspects of the study are summarized in Table 3. No group differences were found for age, weight, height, BMI, or body fat. A post-study questionnaire showed that the double blind procedures were successful, with subjects unable to determine which protein (whey or pea) they were ingesting (Chi square = 1.069, *p* = 0.586).

The physical fitness test data are summarized in Table 4. Significant time effects were found for each test, but exercise-induced changes were modest. None of the interaction *p*-values were significant, indicating that changes in performance during the week did not differ between groups. DOMS increased to high levels 24–48 h post-exercise, but the pattern of change over time did not differ between groups.

The blood chemistry data are summarized in Table 5 and Figure 2 and Figure 3. Groups did not differ significantly pre-exercise for any of these variables. Significant interaction effects were found for AST (aspartate aminotransferase), ALT (alanine aminotransferase), BUN (blood urea nitrogen), albumin, creatine kinase, and myoglobin. The interaction statistic for LDH (lactate dehydrogenase) just missed significance (*p* = 0.054). Post-hoc analysis for AST, ALT, BUN, and LDH did not show any significant differences between groups at any time point (after Bonferonni correction). Post-hoc analysis showed that post-exercise change in creatine kinase and myoglobin was lower in the whey protein but not the pea protein group compared to water during the last 2–3 days of recovery, with no differences between the pea and whey protein groups (Figure 2 and Figure 3). The effect sizes when comparing the reduction in creatine kinase and myoglobin with whey protein compared to water were large (Cohen’s d > 0.80); ingestion of pea protein had an intermediate non-significant effect with small to medium effect sizes (Cohen’s d < 0.50). BUN was higher during recovery in both protein groups, but blood levels did not increase to clinically significant levels. 

## 4. Discussion

The 90-min eccentric exercise protocol induced significant DOMS and a sustained efflux of surrogate biomarkers for muscle damage in 92 untrained males. Despite considerable muscle soreness (peaking 24–48 h post-exercise), the study participants were still able to maintain fitness test performance during recovery (Tuesday through to Friday after the Monday eccentric exercise session), a finding reported by others [27]. Ingestion of whey protein (0.9 g/kg each day, Monday through to Thursday, with 0.3 g/kg on Friday morning) significantly attenuated post-exercise serum creatine kinase and myoglobin biomarkers of muscle damage compared to water only. The effect size was large when comparing whey protein and water groups for creatine kinase and myoglobin after 3 and 4 days of recovery from the Monday exercise session (Cohen’s d > 0.80). Ingestion of pea protein isolate had a smaller effect, and when compared to water, no significant differences were found, with small to medium effect sizes (Cohen’s d < 0.50). No group differences were found when comparing the pea and whey protein groups. Large post-exercise increases were also measured for LDH, AST, and ALT, and the patterns of change during the 5-day period of recovery trended lower for whey protein compared to water, but time point contrasts did not differ significantly between groups (after Bonferroni correction). The exercise bout induced modest increases in CRP after 24 h of recovery, with no significant differences between groups.

Of the various nutritional approaches used as countermeasures for post-exercise muscle soreness and damage, protein and BCAA supplementation has been most widely investigated [41,42]. However, the strategy of using protein and BCAA supplements to alleviate post-exercise DOMS, muscle damage, performance decrements, and inflammation does not have strong literature support. Interpretation has been hampered by weaknesses in study designs [27,28,29,30,31,32,36,41,42,43,44,45]. Study limitations in this area often include small sample sizes, providing low statistical power to evaluate the large interindividual variance typically found for surrogate biomarkers of muscle damage. Randomized crossover trials are not recommended because of the repeat bout effect that causes markedly lower serum levels of creatine kinase and myoglobin after the second bout. Other shortcomings in research designs include a short duration of recovery monitoring and sampling, and the inclusion of trained study participants who experience relatively low levels of post-eccentric exercise DOMS and muscle damage, giving little room for protein supplements to exert a countermeasure effect. For these reasons, we utilized a large number of untrained participants who were randomized into three parallel groups, challenged with a whole-body eccentric bout on Monday morning, and then tested and sampled each morning in an overnight fasted state until Friday morning. 

The protein dosing regimen is also a major consideration, and this study focused on high amounts of supplemental protein split into three doses per day for several days post-exercise. For the first time, pea and whey proteins were compared using double blind procedures and compared to water only. Contrary to our hypothesis, whey but not pea protein attenuated the large post-exercise increase in serum creatine kinase and myoglobin, with no measurable effects of either protein on DOMS, exercise performance measures, or CRP. 

The BCAA profile of pea protein (leucine, isoleucine and valine) is 17.9 g/100 g protein and is 24% below that of whey protein (23.7 g/100 g protein). The average study participant weighed 81 kg and consumed 73 g/day of supplemental protein. Thus, participants in the whey and pea protein groups ingested about 17.3 and 13.1 g of supplemental BCAA per day, respectively. The 4.2 g/day difference in BCAA may have been one factor influencing the efflux of muscle damage markers, perhaps through differences in MPS. 

Food intake during the days preceding and following eccentric exercise could influence the extent of muscle damage and soreness [27]. Limitations of this study were that participants did not record diet food intake in food logs and that dietary intake before and during the 5-day study was not controlled. This was a randomized trial, however, and the levels of dietary protein and BCAA consumed by study participants were more than likely similar between groups. The average young male adult in the U.S. consumes about 103 g of dietary protein each day [38]. Studies measuring dietary BCAA intake are scarce, but limited data indicate levels of about 217 mg/kg per day [46]. Thus, the total average BCAA intake from the diets of our male participants was probably close to 17.6 g per day. The supplemental whey and pea protein added considerably to this amount, and the total intake of BCAA and leucine from both the whey and pea groups should have been enough to support MPS and reduce efflux rates of creatine kinase and myoglobin. Other factors, however, can influence postprandial MPS, and these include differences in protein sources for digestion and amino acid absorption, and plasma amino acid availability for muscle perfusion and MPS [19]. The major amino acid for MPS is leucine, and the 35% lower level of leucine in pea compared to whey protein was likely a key factor explaining our results, especially within the context of the extensive muscle damage experienced in our untrained participants [12]. 

## 5. Conclusions

Taken together, these data support the strategy of using three 0.3 g/kg doses per day of whey protein isolate during several days of recovery from intensive eccentric-based exercise to reduce serum levels of muscle damage biomarkers in untrained males. Supplementation with pea protein compared to water had an intermediate but non-significant effect, with no differences found between pea and whey proteins. The whey protein supplement had no influence on DOMS or exercise performance despite the lowering effect on serum muscle damage biomarkers, a finding that has been reported in other similar investigations [27,28]. Thus exercise-induced leakage of muscle proteins, such as creatine kinase, myoglobin, and LDH, has not been strongly or consistently correlated with muscle performance and soreness [28]. This study focused on a comparison of pea and whey protein (as found in commercial products), with the hypothesis that despite a difference in leucine content, there was still a sufficient amount of leucine even in pea protein to reduce post-exercise damage. This hypothesis was not fully supported by the data. In general, future studies should consider using leucine-fortified pea protein isolate supplements or greater dose volumes to match the leucine and BCAA profile of whey protein isolate [23]. Post-exercise mitigation of serum muscle damage biomarkers may be similar between whey and pea protein supplement groups in comparison to water when leucine and BCAA intake levels are comparable.

## Figures and Tables

**Figure 1 nutrients-12-02382-f001:**
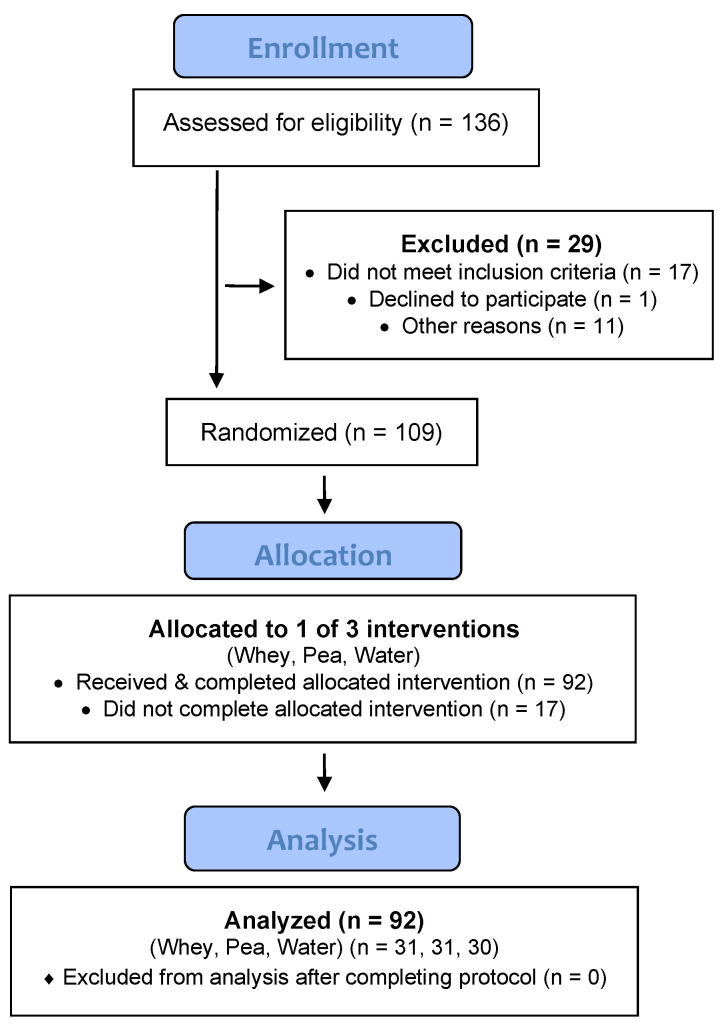
Participant flow diagram.

**Figure 2 nutrients-12-02382-f002:**
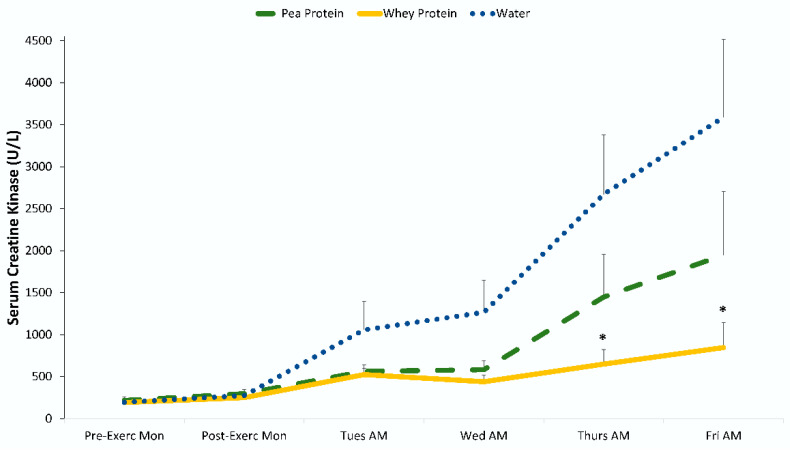
Change in serum creatine kinase. * *p* < 0.01 when comparing change from pre-exercise levels with the whey protein and water-only groups. Cohen’s d = 0.83 and 0.82 on Thursday and Friday mornings, respectively, when comparing whey protein and water groups.

**Figure 3 nutrients-12-02382-f003:**
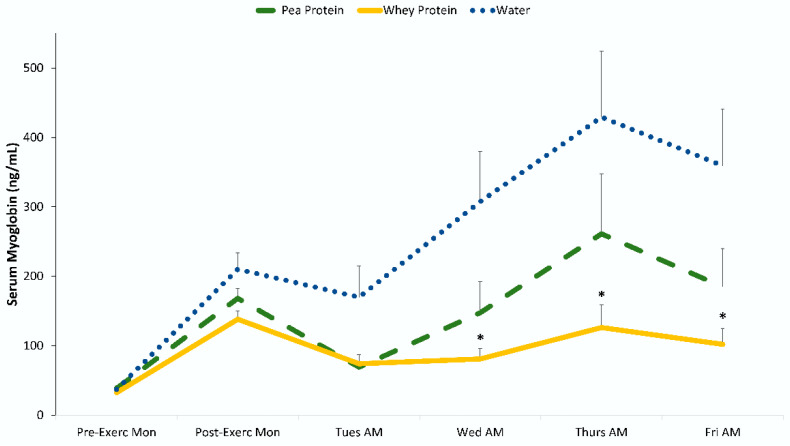
Change in serum myoglobin. * *p* < 0.01 when comparing change from pre-exercise levels with the whey protein and water-only groups. Cohen’s d = 0.93, 0.86, and 0.89 on Wednesday, Thursday, and Friday mornings, respectively, when comparing whey protein and water groups.

**Table 1 nutrients-12-02382-t001:** Amino acid composition (g) for 100 g of pea protein or whey protein supplements *.

Grams/100 g Protein	Pea Protein Isolate	Whey Protein Isolate
**Branched Chain Amino Acids**		
Isoleucine	4.7	5.6
Leucine	8.2	12.7
Valine	5.0	5.4
**Other Amino Acids**		
Alanine	4.3	4.9
Arginine	8.7	2.4
Aspartic acid	11.5	11.4
Cystine	0.1	2.8
Glutamic acid	16.7	16.1
Glycine	4.0	1.7
Histidine	2.5	2.0
Lysine	7.1	10.2
Methionine	1.1	2.3
Phenylalanine	5.5	3.5
Proline	4.3	4.7
Serine	5.1	3.3
Threonine	3.8	4.7
Tryptophan	0.1	2.9
Tyrosine	3.8	3.6

* Sources: USDA National Nutrient Database for Standard Reference and USDEC Reference Manual for U.S. Whey and Lactose Products; Roquette, France.

**Table 2 nutrients-12-02382-t002:** Eccentric exercise protocol.

Dumbbell Incline Bench Press	3 sets, 5 eccentric reps, 23 kg (adjustment as needed). Rest intervals between sets, 30–45 s.
Bench Press	3 sets, 20 s to fatigue, 43 kg, then 3 sets, 3 eccentric reps with concentric assist. Rest intervals, 1 min.
Supine Medicine Ball Catch and Throw	3 sets, 20 s, explosive catch and throw of 6.4 kg medicine ball, with participant in a supine position on floor with knees bent. Rest intervals, 30 s.
Bent Arm Hangs	4 sets, 90° bent arm hang to fatigue. Rest intervals, 30 s.
Eccentric Lat Pulls	4 sets, 8 reps, lat pulls, weight adjusted for eccentric focus. Rest intervals, 1 min.
Downhill Treadmill Run	3 sets, 2 min, 13.7 km/hour, 10% decline. Rest intervals, 1 min.
Drop Jumps with Rebound Vertical Jumps	2 sets, 10 reps, drop jumps from a 41 cm box with explosive vertical jumps. Rest intervals, 30 s.
Explosive Tuck Jumps	3 sets, 20 s, continuous explosive tuck jumps. Rest intervals, 30 s.
Eccentric Leg Extensions	3 sets, 8 reps, each leg, maximal effort against applied resistance for leg extension in 5 s. Rest intervals, 30 s.
Eccentric Leg Curls	2 sets, 8 reps, each leg, maximal effort against applied resistance for leg curl in 5 s, prone position. Rest intervals, 30 s.
Vertical Skips	3 sets, 30 s, skip in place as high as possible. Rest intervals, 1 min.
Split Leg Squats	3 sets, 15 reps, each leg. Rest intervals, 30 secs.
Shrug Walk	Walk 0.53 km on a treadmill with 4.5 kg dumbbells in each hand and a shoulder shrug every two steps.
Bottle Shakers	3 sets, 15 s, isometric abdominal curl with medicine ball (6.4 kg) twisting side to side. Rest intervals, 30 s.
Abdominal Crunches	3 sets, 20 s, supine position with legs bent, repeatedly reaching forward 10 cm. Rest intervals, 30 s.
Plank	2 sets, 45 s, prone position on toes and elbows. Rest intervals, 1 min.

**Table 3 nutrients-12-02382-t003:** Characteristics of the study participants.

Variable	Pea Protein (*n* = 31)	Whey Protein (*n* = 31)	Water (*n* = 30)
Age (years)	37.3 ± 1.6	40.3 ± 1.7	38.1 ± 1.9
Weight (kg)	81.5 ± 1.4	81.5 ± 1.4	80.3 ± 1.8
Height (cm)	179 ± 1.2	178 ± 1.0	179 ± 1.2
BMI (kg/m^2^)	25.4 ± 0.4	25.8 ± 0.4	25.0 ± 0.5
Body Fat (%)	21.1 ± 1.1	20.4 ± 1.0	21.2 ± 1.1

**Table 4 nutrients-12-02382-t004:** Physical fitness test data by supplement group (pea protein, *n* = 31; whey protein, *n* = 31; water, *n* = 30).

Performance Measure	Group	Mon PreEx	Mon PostEx	Tuesday	Wednesday	Thursday	Friday	Time; Treatment; Interaction *p* Values
Vertical Jump (in)	Pea	19.2 ± 0.7	19.6 ± 0.8	19.1 ± 0.8	19.1 ± 0.7	19.5 ± 0.8	19.8 ± 0.8	<0.001; 0.390; 0.723
	Whey	18.0 ± 0.6	18.5 ± 0.6	18.1 ± 0.7	18.3 ± 0.7	18.5 ± 0.7	18.9 ± 0.7	
	Water	18.2 ± 0.4	18.2 ± 0.5	18.4 ± 0.4	17.9 ± 0.5	18.4 ± 0.5	18.7 ± 0.4	
Leg/Back Strength (kg/kg body mass)	Pea	1.64 ± 0.08	1.55 ± 0.09	1.64 ± 0.07	1.63 ± 0.07	1.68 ± 0.07	1.74 ± 0.07	0.002; 0.481; 0.137
	Whey	1.64 ± 0.10	1.68 ± 0.10	1.58 ± 0.08	1.62 ± 0.08	1.68 ± 0.09	1.69 ± 0.08	
	Water	1.56 ± 0.07	1.52 ± 0.08	1.51 ± 0.06	1.50 ± 0.07	1.52 ± 0.07	1.62 ± 0.07	
Bench Press (repetitions)	Pea	14.0 ± 1.3	10.9 ± 1.5	12.8 ± 1.6	13.5 ± 1.7	13.9 ± 1.7	14.4 ± 1.7	<0.001; 0.141; 0.251
	Whey	12.9 ± 1.1	9.48 ± 1.22	11.9 ± 1.3	12.8 ± 1.3	13.2 ± 1.3	14.3 ± 1.3	
	Water	11.6 ± 1.2	7.13 ± 1.16	8.80 ± 1.29	9.23 ± 1.21	10.1 ± 1.3	11.1 ± 1.4	
Wingate Peak Power (watts/kg)	Pea	8.36 ± 0.30	7.54 ± 0.35	8.12 ± 0.26	8.05 ± 0.31	8.29 ± 0.30	8.41 ± 0.29	<0.001; 0.464; 0.321
	Whey	7.92 ± 0.25	7.27 ± 0.28	7.72 ± 0.21	7.82 ± 0.23	7.91 ± 0.22	7.99 ± 0.24	
	Water	8.54 ± 0.33	7.80 ± 0.38	8.45 ± 0.29	8.16 ± 0.33	8.26 ± 0.33	8.16 ± 0.32	
Wingate Mean Power (watts/kg)	Pea	6.50 ± 0.16	5.84 ± 0.22	6.37 ± 0.15	6.32 ± 0.18	6.44 ± 0.16	6.51 ± 0.15	<0.001; 0.926; 0.588
	Whey	6.52 ± 0.17	5.96 ± 0.21	6.38 ± 0.18	6.45 ± 0.19	6.61 ± 0.18	6.56 ± 0.18	
	Water	6.55 ± 0.23	6.09 ± 0.32	6.55 ± 0.24	6.41 ± 0.25	6.51 ± 0.26	6.50 ± 0.25	
Delayed Onset Muscle Soreness	Pea	1.45 ± 0.11	3.69 ± 0.40	6.23 ± 0.27	6.00 ± 0.25	4.06 ± 0.29	2.45 ± 0.25	<0.001; 0.283; 0.210
	Whey	1.42 ± 0.13	4.08 ± 0.34	6.19 ± 0.30	5.50 ± 0.31	3.66 ± 0.25	2.42 ± 0.20	
	Water	1.22 ± 0.07	4.25 ± 0.40	6.72 ± 0.33	6.37 ± 0.47	4.68 ± 0.45	3.03 ± 0.38	

Measurements conducted pre- and post-eccentric exercise (PreEx, PostEx) on Monday, and each morning through 4 days of recovery (Tuesday through to Friday).

**Table 5 nutrients-12-02382-t005:** Blood chemistry data by group.

Blood Measure	Group	Mon PreEx	Mon PostEx	Tuesday	Wednesday	Thursday	Friday	Time; Treatment; Interaction *p* Values
AST (IU/L)	Pea	24.8 ± 1.7	27.8 ± 1.8	32.3 ± 1.9	30.7 ± 1.6	40.4 ± 6.8	49.5 ± 10.4	<0.001; 0.018; 0.025
	Whey	22.1 ± 1.1	24.6 ± 6.7	30.5 ± 2.1	29.6 ± 2.3	30.3 ± 2.8	33.4 ± 4.1	
	Water	22.7 ± 1.2	26.3 ± 1.3	43.8 ± 7.1	47.8 ± 9.1	63.7 ± 13.0	79.9 ± 17.3	
ALT (IU/L)	Pea	22.1 ± 1.2	23.5 ± 1.3	22.3 ± 1.2	23.9 ± 1.2	27.0 ± 1.9	30.6 ± 3.4	<0.001; 0.196; 0.029
	Whey	22.2 ± 1.8	23.7 ± 1.9	23.2 ± 1.7	25.7 ± 2.1	25.8 ± 1.8	26.9 ± 1.9	
	Water	22.1 ± 1.5	24.2 ± 1.4	25.2 ± 1.7	27.9 ± 2.2	32.9 ± 3.3	40.1 ± 4.9	
BUN (mg/dL)	Pea	13.6 ± 0.5	14.7 ± 0.5 *	15.5 ± 0.6 *	15.3 ± 0.6 *	15.1 ± 0.7 *	15.3 ± 0.5	0.015; <0.001 0.001
	Whey	15.8 ± 0.5	17.6 ± 0.5 *	17.9 ± 0.6 *	17.9 ± 0.5 *	17.3 ± 0.6 *	18.2 ± 0.4 *	
	Water	14.4 ± 0.6	14.7 ± 0.6	13.5 ± 0.5	12.9 ± 0.6	13.2 ± 0.6	13.8 ± 0.7	
CRP (mg/L)	Pea	1.08 ± 0.25	1.11 ± 0.26	1.84 ± 0.42	1.71 ± 0.36	1.37 ± 0.34	1.34 ± 0.39	<0.001; 0.133 0.549
	Whey	0.64 ± 0.16	0.69 ± 0.16	1.10 ± 0.23	0.89 ± 0.18	0.74 ± 0.13	0.71 ± 0.11	
	Water	0.84 ± 0.14	0.92 ± 0.16	2.09 ± 0.44	1.69 ± 0.41	1.30 ± 0.31	1.30 ± 0.35	
Albumin (g/dL)	Pea	4.55 ± 0.04	4.81 ± 0.04	4.52 ± 0.04	4.51 ± 0.05	4.51 ± 0.05	4.56 ± 0.05	<0.001; 0.057 0.042
	Whey	4.65 ± 0.04	4.85 ± 0.04	4.66 ± 0.04	4.59 ± 0.04	4.56 ± 0.04	4.65 ± 0.05	
	Water	4.68 ± 0.03	5.01 ± 0.05	4.58 ± 0.04	4.59 ± 0.04	4.60 ± 0.03	4.70 ± 0.04	
Glucose (mg/dL)	Pea	92.4 ± 1.6	104 ± 5.2	93.8 ± 1.7	94.5 ± 1.5	93.2 ± 1.5	94.1 ± 1.5	0.002; 0.908 0.068
	Whey	95.5 ± 1.3	95.3 ± 2.2	94.7 ± 1.9	95.4 ± 1.5	96.4 ± 1.4	95.5 ± 1.6	
	Water	94.6 ± 1.3	103 ± 3.8	94.3 ± 1.0	92.7 ± 0.9	92.6 ± 1.3	92.2 ± 1.5	
LDH (IU/L)	Pea	162 ± 4.7	185 ± 4.5	172 ± 4.6	172 ± 4.6	193 ± 14.6	204 ± 20.4	<0.001; 0.044; 0.054
	Whey	164 ± 5.1	188 ± 5.6	174 ± 5.5	173 ± 5.8	174 ± 7.2	181 ± 9.5	
	Water	167 ± 3.8	198 ± 5.3	185 ± 7.2	194 ± 10.2	235 ± 22.5	251 ± 28.3	
Estimated GFR (mL/min/1.73)	Pea	90.1 ± 2.4	72.6 ± 2.5	87.0 ± 3.2	90.7 ± 2.2	93.7 ± 2.5	91.9 ± 2.2	<0.001; 0.930; 0.316
	Whey	90.6 ± 2.5	75.8 ± 2.4	90.5 ± 2.4	94.6 ± 2.2	95.7 ± 2.3	93.7 ± 2.0	
	Water	99.0 ± 2.5	81.5 ± 2.3	93.8 ± 2.9	99.1 ± 2.8	101 ± 3.0	99.3 ± 3.4	

AST (aspartate aminotransferase), ALT (alanine aminotransferase), BUN (blood urea nitrogen), CRP (C-reactive protein), LDH (lactate dehydrogenase), GFR (glomerular filtration rate). Measured pre- and post-eccentric exercise on Monday, and each morning through 4 days of recovery (Tuesday through to Friday). * *p* ≤ 0.01 vs. water Bonferroni adjustment.

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
