# Peer review of "Effects of Whey and Pea Protein Supplementation on Post-Eccentric Exercise Muscle Damage: A Randomized Trial"

_nutrients, 2020, doi:10.3390/nu12082382_

Round 1
Reviewer 1 Report
General Comments
This article investigated the effect of whey and pea protein supplementation on post-eccentric exercise muscle damage, inflammation, delayed onset of muscle soreness (DOMS), and physical fitness test performance during a 5-day period in non-athletic, non-obese males. This is potentially interesting research; however, without a record of their food intake, the authors cannot examine their hypothesis. Since total protein intake during the day affects the maintenance of the muscle mass, the authors should indicate a food intake record of their participants. Especially the participants can intake leucine by themselves, thus, the significance of this study is minimal without the record. The inclusion of such evidence would have made this manuscript publishable regardless of if the hypothesis would be confirmed or rejected. The specific comments follow.
Specific Comments
Methods;
- Why the authors did not measure the range of motion and circumference of the exercised muscles? Since those are the important indexes of DOMS used in various studies, the data will strengthen this study.
Results;
- How much protein did the participants take one day? It can directly affect outcomes in this study so please indicate the data.
Discussion;
- Why did the authors use pea protein in this study? The pea protein originally had lower leucine concentration than whey protein and had smaller effects on the makers of the DOMS. The reviewers could not understand the advantage of using pea protein in this study. Please explain and discuss that.
- Although the results of fitness tests already got back to the baseline on the next day of eccentric exercise, the blood markers were still higher than baseline even the end of the experiment period. The readers do not understand this discrepancies. Please explain more the relationship among these blood makers, fitness test outcomes, and DOMS.
Author Response
REVIEWER #1
General Comments
This article investigated the effect of whey and pea protein supplementation on post-eccentric exercise muscle damage, inflammation, delayed onset of muscle soreness (DOMS), and physical fitness test performance during a 5-day period in non-athletic, non-obese males. This is potentially interesting research; however, without a record of their food intake, the authors cannot examine their hypothesis. Since total protein intake during the day affects the maintenance of the muscle mass, the authors should indicate a food intake record of their participants. Especially the participants can intake leucine by themselves, thus, the significance of this study is minimal without the record. The inclusion of such evidence would have made this manuscript publishable regardless of if the hypothesis would be confirmed or rejected. The specific comments follow.
RESPONSE: Thank you for taking the time and effort to review our paper within your busy schedule. We addressed your comments below.
Specific Comments
Methods;
- Why the authors did not measure the range of motion and circumference of the exercised muscles? Since those are the important indexes of DOMS used in various studies, the data will strengthen this study.
RESPONSE: There are many different outcome measures for DOMS. We used a self-reported scale, and 4 fitness tests to determine effects on performance. We agree that the two measures you mentioned would have added to the value of the data, but our primary focus was on muscle damage markers.
Results;
- How much protein did the participants take one day? It can directly affect outcomes in this study so please indicate the data.
RESPONSE: We discuss protein intake (2.15 g/kg per day) in the last paragraph of the introduction, and the last paragraph of the main discussion section.
Discussion;
- Why did the authors use pea protein in this study? The pea protein originally had lower leucine concentration than whey protein and had smaller effects on the makers of the DOMS. The reviewers could not understand the advantage of using pea protein in this study. Please explain and discuss that.
RESPONSE: The introduction (4th paragraph) reviews the rationale for using pea protein.
"Intake of plant protein has increased during the past two decades in part due to environmental advantages and the linkage to improved health and decreased all-cause mortality [33,34]…Table 1 compares the amino acid composition for 100 g of the pea protein and whey protein supplements used in this trial [35]. One study showed that 50 g/d doses of pea or whey protein during a 12-week resistance training period resulted in similar increases in muscle thickness relative to placebo, [35]. These data suggest that the leucine content of pea protein, although 24% below that of whey protein, is at a sufficient level to support exercise-induced improvements in muscle hypertrophy, but this interpretation has been challenged as spurious [12]."
- Although the results of fitness tests already got back to the baseline on the next day of eccentric exercise, the blood markers were still higher than baseline even the end of the experiment period. The readers do not understand this discrepancies. Please explain more the relationship among these blood makers, fitness test outcomes, and DOMS.
RESPONSE: We added further discussion of this in the conclusion section (in response to your recommendation).
" The whey protein supplement had no influence on DOMS or exercise performance despite the lowering effect on serum muscle damage biomarkers, a finding that has been reported in other similar investigations [27,28]. Thus exercise-induced leakage of muscle proteins such as creatine kinase, myoglobin, and LDH has not been strongly or consistently correlated with muscle performance and soreness [28]."
-----------------------------
Reviewer 2 Report
The authors have investigated the impact of whey and pea protein supplementation on indirect markers of post-exercise muscle damage. The protein supplementation protocol consistent out of 3 strategically timed doses pre-exercise, post-exercise, and pre-sleep (0.3 g/kg each). The authors observed no benefit of exercise performance tests, but whey protein appeared to reduce some of the blood markers of muscle damage. The study is well-designed to pick up a potential benefit (e.g. decent sample size, high total amount of protein supplementation, strategic timing, population with room for improvement, multiday protocol etc). However, I have some comments regarding the writing and data interpretation I would like the authors to address.
Line 21: I miss 1-2 lines of background information in the abstract.
Line 35: it cannot be concluded from the current experiment that the intermediate effect of pea protein is due to the lower leucine content. In addition, this new information/explanation should not be introduced in the last sentence (the conclusion).
Line 49: I would a bit more conservative and suggest that leucine appears to be a strong determinant (instead of the strongest).
Line 51: Ref 13 is used in several places to support various statements. However, ref 13 is a relatively broad review covering various topics (which would make the reader search in the article) and many topics are not addressed in detail. I would be better to replace or add a more specific citation for the statement here (e.g. PMID: 23459753)
Line 54: Given your supplemental timing protocol, it may be worthwhile to indicate that this statement is also true for pre-sleep protein ingestion (e.g. PMID: 28536184)
Line 71: brand names should not be mentioned, except once in the methods section. Details about the study product (e.g. its composition) should also be presented in the methods section.
Line 74-78: this conclusion that the leucine content of pea protein is at a sufficient level to support exercise-induced improvements in muscle hypertrophy is incorrect. The provided reference found no significant difference between protein supplementation (whey or pea) compared to placebo.
Line 78: the purpose of this sentence about oat protein is not readily clear to me. I think its purpose is to suggest that plant-based protein supplementation can potentially be effective at reducing muscle damage? If so, make this more clear for the reader. Is that the study the only work examining the impact of plant-based protein supplementation on muscle damage? If so, you can state this and use it as a set up why pea protein was used in the present study.
Line 86: I miss a rationale before introducing the study. Lines 254-266 from the discussion does a good job introducing why this study is worthwhile and would fit better in the introduction when compared to the discussion.
Introduction: After reading the introduction, it was not readily clear to me why pea protein was chosen. I understand it was of interest to the sponsor, but more rationale should be given to the reader. For example, is the amino acid composition (high leucine, BCAA, and/or EAA content) and/or digestibility score, or a combination of both (DIAAS score; PMID: 28382889) higher when compared to (most) other plant sources?
Line 91: “the average dietary…..”
This does not below in the introduction. This calculation is also not as simple, as habitual protein intake may decrease as a result of protein supplementation.
Line 105: The study described as NCT03448328 does not match the current manuscript (e.g. it mentions apple juice and does not mention the third supplemental dose.
Line 208: Why is legBack expressed per kg BM?
Figure 2 and 3: when printing the paper black white, it’s almost impossible to identify the treatments. Please either add different icons to the different treatment and/or change the line pattern for the different treatments (e.g. a dashed line for pea, solid line for whey, etc). If the authors wish to keep the colours, I would make them more instinctive: use the green colour for the pea protein, and the blue color for the water treatment.
General. It is not clear to me why blood markers are considered the main outcome. They are considered as poor markers of muscle damage (e.g. PMID: 11252462). Functional outcomes in the performance tests seem more relevant? The current data appear to suggest that protein supplementation in general is not effective at reducing muscle damage.
Optional: perhaps cluster the two proteins together to investigate whether protein reduces your blood and functional markers of muscle damage.
Author Response
Reviewer #2
Comments and Suggestions for Authors
The authors have investigated the impact of whey and pea protein supplementation on indirect markers of post-exercise muscle damage. The protein supplementation protocol consistent out of 3 strategically timed doses pre-exercise, post-exercise, and pre-sleep (0.3 g/kg each). The authors observed no benefit of exercise performance tests, but whey protein appeared to reduce some of the blood markers of muscle damage. The study is well-designed to pick up a potential benefit (e.g. decent sample size, high total amount of protein supplementation, strategic timing, population with room for improvement, multiday protocol etc). However, I have some comments regarding the writing and data interpretation I would like the authors to address.
RESPONSE: Thank you for taking the time and effort to review our paper within your busy schedule.
Line 21: I miss 1-2 lines of background information in the abstract.
RESPONSE: The complete abstract is available with the paper.
Line 35: it cannot be concluded from the current experiment that the intermediate effect of pea protein is due to the lower leucine content. In addition, this new information/explanation should not be introduced in the last sentence (the conclusion).
RESPONSE: Dropped this from the abstract, in response to your comment.
Line 49: I would a bit more conservative and suggest that leucine appears to be a strong determinant (instead of the strongest).
RESPONSE: Changed as you recommended.
Line 51: Ref 13 is used in several places to support various statements. However, ref 13 is a relatively broad review covering various topics (which would make the reader search in the article) and many topics are not addressed in detail. I would be better to replace or add a more specific citation for the statement here (e.g. PMID: 23459753)
RESPONSE: Added this reference [25], as you recommended.
Line 54: Given your supplemental timing protocol, it may be worthwhile to indicate that this statement is also true for pre-sleep protein ingestion (e.g. PMID: 28536184)
RESPONSE: Added this phrase, as you recommended.
Line 71: brand names should not be mentioned, except once in the methods section. Details about the study product (e.g. its composition) should also be presented in the methods section.
RESPONSE: Removed from the introduction, as you recommended.
Line 74-78: this conclusion that the leucine content of pea protein is at a sufficient level to support exercise-induced improvements in muscle hypertrophy is incorrect. The provided reference found no significant difference between protein supplementation (whey or pea) compared to placebo.
RESPONSE: Dropped this statement.
Line 78: the purpose of this sentence about oat protein is not readily clear to me. I think its purpose is to suggest that plant-based protein supplementation can potentially be effective at reducing muscle damage? If so, make this more clear for the reader. Is that the study the only work examining the impact of plant-based protein supplementation on muscle damage? If so, you can state this and use it as a set up why pea protein was used in the present study.
RESPONSE: Tried to make this more clear.
" Few studies have evaluated the effect of plant protein ingestion on exercise-induced muscle damage. Supplementation with oat protein (25 g/day, 18 days) in one study was linked to a decrease in delayed-onset of muscle soreness (DOMS), serum creatine kinase and myoglobin, and serum C-reactive protein (CRP) after an intense downhill treadmill run [36]."
Line 86: I miss a rationale before introducing the study. Lines 254-266 from the discussion does a good job introducing why this study is worthwhile and would fit better in the introduction when compared to the discussion…. Introduction: After reading the introduction, it was not readily clear to me why pea protein was chosen. I understand it was of interest to the sponsor, but more rationale should be given to the reader. For example, is the amino acid composition (high leucine, BCAA, and/or EAA content) and/or digestibility score, or a combination of both (DIAAS score; PMID: 28382889) higher when compared to (most) other plant sources?
RESPONSE: We left the discussion section as is because we felt those statements were needed there. In response to your comment, we strengthened the rationale statement in paragraph 3 and the last paragraph of the introduction.
"We utilized a randomized, parallel group design, and emphasized high amounts of supplemental protein split into three doses per day for several days post-exercise. The three daily acute pea and whey protein doses (0.3 g/k) were designed to provide adequate amounts of leucine within the recommended range of 700-3,000 mg [13]. Thus, we hypothesized.."
We used pea protein because of the current interest in plant-based protein by individuals engaging in resistance exercise (paragraph 4, introduction).
Line 91: “the average dietary…..”
This does not below in the introduction. This calculation is also not as simple, as habitual protein intake may decrease as a result of protein supplementation.
RESPONSE: Moved this to the methods.
Line 105: The study described as NCT03448328 does not match the current manuscript (e.g. it mentions apple juice and does not mention the third supplemental dose.
RESPONSE: We switched to water only, and added a third dose in response to some pilot data.
Line 208: Why is legBack expressed per kg BM?
RESPONSE: To adjust for the wide range of body masses in our participants.
Figure 2 and 3: when printing the paper black white, it’s almost impossible to identify the treatments. Please either add different icons to the different treatment and/or change the line pattern for the different treatments (e.g. a dashed line for pea, solid line for whey, etc). If the authors wish to keep the colours, I would make them more instinctive: use the green colour for the pea protein, and the blue color for the water treatment.
RESPONSE: Changed Figures 2 and 3 as recommended.
General. It is not clear to me why blood markers are considered the main outcome. They are considered as poor markers of muscle damage (e.g. PMID: 11252462). Functional outcomes in the performance tests seem more relevant? The current data appear to suggest that protein supplementation in general is not effective at reducing muscle damage.
RESPONSE: Reviews in this area emphasize that exercise-induced changes in muscle damage biomarkers, performance tests, and DOMS are often unrelated. We chose the more objective serum muscle damage biomarkers as our primary outcomes, with DOMS and performance tests as secondary outcomes.
Optional: perhaps cluster the two proteins together to investigate whether protein reduces your blood and functional markers of muscle damage.
RESPONSE: We feel the comparison of pea and whey proteins is the novel aspect of our study.
Reviewer 3 Report
The manuscript by Nieman et al examines the effects of short term (5 days) whey and pea protein supplementations on muscle damage and fitness test performance in general healthy population age 18-55 years old. The strength of this study is that authors recruited participants from an apparently healthy population and conducted a double blind clinical trial. The comparison was made 1) within-subjects for the time course of exercise and supplementation analysis, 2) among the groups with different types of supplements (whey, pea proteins and water). Despite the strengths, the hypothesis is not novel and the results seem to be somewhat expected. However, comparing the efficacy of decreased muscle damage biomarkers between dairy- and plant- based protein supplementation may be somewhat interesting. Overall, the data may be attractive to this field of the scientific community, but I have several concerns with the current manuscript.
Major:
Introduction:
Line 91
Remove these last sentences or move them to the discussion. They don’t seem to belong in the introduction.
Methods:
How did the authors control daily food intake? How did they measure physical activity levels other than during the exercise session? If it was not controlled, these should be stated.
Line 146-157
The flow chart and diet composition tables are well described, but exercise protocol description is difficult to follow. Change this part to something like Table X describes the exercise protocol as described previously (40)
Line 158- 159
The assessment of the DOMS rating immediately after the exercise does not make sense. It is “Delayed”.
Line 172
Provide more details of the sample preparation. It sounds like a serum, but it is not clear if it was serum or plasma.
Line 185
Why is a “between subjects comparison” performed by a repeated ANOVA? Time course should be repeated, but the group difference should be a simple ANOVA. The wording of the statistics description is confusing and the statistics section needs to be reviewed by a statistician. What is the purpose of the time course analysis anyway?
Results:
Table 3 is overwhelmed with numbers as a table. First, what is the purpose of pre/post-exercise data on Monday? Also, why do you need perform the fitness test every day? Are the authors trying to validate the sensitivity of exercise tests by a tester? 2nd, how can participants provide DOMS scores before exercise on Monday (again “delayed onset”)? Finally, what do authors want to convey by providing both Wingate Peak and Mean power data? It is not clear what message the authors is trying to deliver with these data.
Table 4 (time course differences of serum biomarkers) is interesting. However, some of the data supporting or denying their hypothesis in this table should be provided like figure 2 and 3. As of right now, the authors simply present the numbers and it is very difficult to understand the message the authors want to deliver.
Based on table 2, the physical characteristics of the participants appear to be similar. However, metabolic panel data and inflammatory biomarkers (table 4) suggest that there seems to be some differences among the group. For example, CRP data in PreEx data from the whey group is much lower than in the other groups. This should be discussed in the discussion.
Discussion/Conclusions:
Line277-279
I do agree that the major weakness of this study is that diet intake and physical activity were not measured or controlled. It is an over-speculation that protein intake from diet among the groups is likely to be the same. Indeed, as mentioned previously, there seem to be some differences among the groups based on the biochemical data. Specifically, estimated GFR levels are lower in protein supplement groups (roughly ~10% lower than the water group). The authors did not explain the calculation methods that should be explained in the methods section, but I expect that creatinine values were lower in these 2 groups. Given BUN values are similar among the groups, I have some concerns about pre-renal issues. Based on the data, it is possible that the hydration status in these 2 groups were different.
It may be ok to state the author’s opinion about the differences between whey and pea protein supplementation in the discussion section, but I do not agree with the conclusion in the abstract saying that the pea protein supplement had an intermediate effect in part due to the 24% lower leucine amino acid content. This study is not designed to make this conclusion. The differences between the whey and pea protein supplementations they observed are not known. This sentence in the abstract should be modified.
Author Response
Reviewer #3
Comments and Suggestions for Authors
The manuscript by Nieman et al examines the effects of short term (5 days) whey and pea protein supplementations on muscle damage and fitness test performance in general healthy population age 18-55 years old. The strength of this study is that authors recruited participants from an apparently healthy population and conducted a double blind clinical trial. The comparison was made 1) within-subjects for the time course of exercise and supplementation analysis, 2) among the groups with different types of supplements (whey, pea proteins and water). Despite the strengths, the hypothesis is not novel and the results seem to be somewhat expected. However, comparing the efficacy of decreased muscle damage biomarkers between dairy- and plant- based protein supplementation may be somewhat interesting. Overall, the data may be attractive to this field of the scientific community, but I have several concerns with the current manuscript.
RESPONSE: Thank you for taking the time and effort to review our paper within your busy schedule.
Major:
Introduction:
Line 91
Remove these last sentences or move them to the discussion. They don’t seem to belong in the introduction.
RESPONSE: Moved to the methods.
Methods:
How did the authors control daily food intake? How did they measure physical activity levels other than during the exercise session? If it was not controlled, these should be stated.
RESPONSE: We did not control or measure food intake or physical activity during this 5-day study (added this statement to the methods in response to your comment, section 2.2).
Line 146-157
The flow chart and diet composition tables are well described, but exercise protocol description is difficult to follow. Change this part to something like Table X describes the exercise protocol as described previously (40)
RESPONSE: Added a table (Table 2) with details about the eccentric exercise protocol in response to your comment.
Line 158- 159
The assessment of the DOMS rating immediately after the exercise does not make sense. It is “Delayed”.
RESPONSE: Changed to muscle soreness rating.
Line 172
Provide more details of the sample preparation. It sounds like a serum, but it is not clear if it was serum or plasma.
RESPONSE: Provided more information for serum preparation.
Line 185
Why is a “between subjects comparison” performed by a repeated ANOVA? Time course should be repeated, but the group difference should be a simple ANOVA. The wording of the statistics description is confusing and the statistics section needs to be reviewed by a statistician. What is the purpose of the time course analysis anyway?
RESPONSE: This section is accurate as stated. DCN teaches statistics and performed the analysis. This statistical analysis approach has been used in many prior studies published by our research group.
Results:
Table 3 is overwhelmed with numbers as a table. First, what is the purpose of pre/post-exercise data on Monday? Also, why do you need perform the fitness test every day? Are the authors trying to validate the sensitivity of exercise tests by a tester? 2nd, how can participants provide DOMS scores before exercise on Monday (again “delayed onset”)? Finally, what do authors want to convey by providing both Wingate Peak and Mean power data? It is not clear what message the authors is trying to deliver with these data.
RESPONSE: We feel that the data in this table are needed, and the research design is adequately explained in the methods section. The pre-post-test data on Monday provided the immediate effect of the bout on all outcome measures. Few studies have followed participants for four recovery days, and this study provides important data on the time course of recovery.
Table 4 (time course differences of serum biomarkers) is interesting. However, some of the data supporting or denying their hypothesis in this table should be provided like figure 2 and 3. As of right now, the authors simply present the numbers and it is very difficult to understand the message the authors want to deliver.
RESPONSE: Figures 2 and 3 summarize the primary outcome measures for the study (and the most interesting data). The tables summarize the secondary outcome measures and we do not feel that figures are needed for those data because interaction effects or between group post-hoc analyses are largely insignificant. BUN had some group differences that we address in the discussion, but we concluded that increases with protein intake did not increase BUN above the normal clinical range.
Based on table 2, the physical characteristics of the participants appear to be similar. However, metabolic panel data and inflammatory biomarkers (table 4) suggest that there seems to be some differences among the group. For example, CRP data in PreEx data from the whey group is much lower than in the other groups. This should be discussed in the discussion.
RESPONSE: There were no pre-exercise differences between groups for any of the outcome measures. We added a statement in the results section to address your concern.
Discussion/Conclusions:
Line277-279
I do agree that the major weakness of this study is that diet intake and physical activity were not measured or controlled. It is an over-speculation that protein intake from diet among the groups is likely to be the same. Indeed, as mentioned previously, there seem to be some differences among the groups based on the biochemical data. Specifically, estimated GFR levels are lower in protein supplement groups (roughly ~10% lower than the water group). The authors did not explain the calculation methods that should be explained in the methods section, but I expect that creatinine values were lower in these 2 groups. Given BUN values are similar among the groups, I have some concerns about pre-renal issues. Based on the data, it is possible that the hydration status in these 2 groups were different.
RESPONSE: There were no pre-exercise group differences for any of these variables. We acknowledge the study limitation of not measuring the data. Nonetheless, excellent national data are available to provide an approximation of protein intake for this age group. Also, this is a study that utilized a randomized, parallel group design with a relatively large group of males. Within that context, it is unlikely that groups differed significantly in dietary protein intake or other potential confounders such as physical activity.
It may be ok to state the author’s opinion about the differences between whey and pea protein supplementation in the discussion section, but I do not agree with the conclusion in the abstract saying that the pea protein supplement had an intermediate effect in part due to the 24% lower leucine amino acid content. This study is not designed to make this conclusion. The differences between the whey and pea protein supplementations they observed are not known. This sentence in the abstract should be modified.
RESPONSE: This statement has been removed from the abstract.
Round 2
Reviewer 1 Report
Manuscript Number nutrients-883359
Effects of Whey and Pea Protein Supplementation on Post-Eccentric Exercise Muscle Damage: A Randomized Trial
General Comments
The authors misunderstand the meanings of my comments, thus the reviewer ask them an additional revise. The comments follow.
Specific Comments
Results;
- How much protein did the participants take one day? It can directly affect outcomes in this study so please indicate the data.
RESPONSE: We discuss protein intake (2.15 g/kg per day) in the last paragraph of the introduction, and the last paragraph of the main discussion section.
COMENTS: The reviewer knows the authors provided that information. However, it is not the exact values of each intervention group, right? Unless the authors control the food intake of all participants, it would never happen. My question is how much protein did the participants of each group take one day? Please the data of each group with variations. If the authors do not have the data, they should include this point in their discussion and limitation in the revised manuscript.
Discussion;
- Why did the authors use pea protein in this study? The pea protein originally had lower leucine concentration than whey protein and had smaller effects on the makers of the DOMS. The reviewers could not understand the advantage of using pea protein in this study. Please explain and discuss that.
RESPONSE: The introduction (4 th paragraph) reviews the rationale for using pea protein. "Intake of plant protein has increased during the past two decades in part due to environmental advantages and the linkage to improved health and decreased all-cause mortality [33,34]…Table 1 compares the amino acid composition for 100 g of the pea protein and whey protein supplements used in this trial [35]. One study showed that 50 g/d doses of pea or whey protein during a 12-week resistance training period resulted in similar increases in muscle thickness relative to placebo, [35]. These data suggest that the leucine content of pea protein, although 24% below that of whey protein, is at a sufficient level to support exercise-induced improvements in muscle hypertrophy, but this interpretation has been challenged as spurious [12]."
COMENTS: The reviewer knows the authors provided the rationale for using pea protein in their Introduction. If they used only the pea protein or they compared the effects with the same amount of leucine supplementation (8.2 g/100 g protein) in this study, that makes sense for the reviewer. However, they compared the effects with whey protein which had a different amount of leucine (12.7 g/100 g protein). That is a sort of question on their experimental design.
In other words, as long as reading the introduction, the reviewer thought that the authors would like to focus on the effects of pea protein on muscle damage because they did not mention about whey protein in their introduction. If that is correct, the authors should use whey protein with the same amount of leucine supplementation (8.2 g/100 g protein) as a positive control. If pea protein will have equal or higher effects, the pea protein supplementation will be beneficial as the authors hypothesized. However, they used pea protein and whey protein which had a different amount of leucine (12.7 g/100 g protein); thus, the readers will confuse the reason why the authors used pea protein (8.2 g/100 g protein) in this study.
If the authors would like to focus on both pea protein and whey protein which includes the same amount of total protein, they should revise their introduction in order to understand the rationale for using pea protein and whey protein in this study for the readers.
Author Response
General Comments
The authors misunderstand the meanings of my comments, thus the reviewer ask them an additional revise.The comments follow.
Specific Comments
Results;
- How much protein did the participants take one day? It can directly affect outcomes in this study so please indicate the data.
RESPONSE: We discuss protein intake (2.15 g/kg per day) in the last paragraph of the introduction, and the last paragraph of the main discussion section.
COMMENTS:The reviewer knows the authors provided that information. However, it is not the exact values of each intervention group, right? Unless the authors control the food intake of all participants, it would never happen. My question is how much protein did the participants of each group take one day? Please the data of each group with variations. If the authors do not have the data, they should include this point in their discussion and limitation in the revised manuscript.
RESPONSE:
We appreciate your comments and your recommendations to improve the paper. We previously added this statement in the methods lines 125 to 129:
The average dietary protein intake for the age group selected for this study (ages 18 to 55 years) is 103±4.3 g or approximately 1.25 g/kg per day for non-obese males [37]. Thus, dietary intake for protein combined with the supplement was calculated at the high but safe level of 2.15 g/kg per day, well below the 3.5 g/kg tolerable upper level [38].
We edited this statement (in response to your comment) to ensure readers know this is an estimation based on national data.
The average dietary protein intake using national data for the age group selected for this study (ages 18 to 55 years) is 103±4.3 g or approximately 1.25 g/kg per day for non-obese males [37]. Thus, dietary intake for protein combined with the supplement was calculated at the high but safe level of 2.15 g/kg per day, well below the 3.5 g/kg tolerable upper level [38].
In the discussion, lines 283 to 286, we previously added this statement:
Food intake during the days preceding and following eccentric exercise could influence the extent of muscle damage and soreness [27]. Limitations of this study were that participants did not record diet food intake in food logs and that dietary intake before and during the 5-day study was not controlled.
These statements clearly indicate that we did not measure dietary or protein intake, and we list this as a limitation. As emphasized in the discussion (lines 286 to 288), in studies using randomized groups, intake levels of dietary protein and BCAA are similar. In studies that do not include randomized control groups, measurement of food intake is more critical because it can confound results. Obviously, this is the reason for using randomized control groups (i.e., to control for confounding effects from unmeasured factors).
Discussion;
- Why did the authors use pea protein in this study? The pea protein originally had lower leucine concentration than whey protein and had smaller effects on the makers of the DOMS. The reviewers could not understand the advantage of using pea protein in this study. Please explain and discuss that.
RESPONSE: The introduction (4 th paragraph) reviews the rationale for using pea protein."Intake of plant protein has increased during the past two decades in part due to environmental advantages and the linkage to improved health and decreased all-cause mortality [33,34]…Table 1 compares the amino acid composition for 100 g of the pea proteinand whey protein supplements used in this trial [35]. One study showed that 50 g/d doses of pea or whey protein during a 12-week resistance training period resulted in similar increases in muscle thickness relative to placebo, [35]. These data suggest that the leucine content of pea protein, although 24% below that of whey protein, is at a sufficient level to support exercise-induced improvements in muscle hypertrophy, but this interpretation has been challenged as spurious [12]."
COMMENTS:The reviewer knows the authors provided the rationale for using pea protein in their Introduction. If they used only the pea protein or they compared the effects with the same amount of leucine supplementation (8.2 g/100 g protein) in this study, that makes sense for the reviewer. However, they compared the effects with whey protein which had a different amount of leucine (12.7 g/100 g protein). That is a sort of question on their experimental design.
In other words, as long as reading the introduction, the reviewer thought that the authors would like to focus on the effects of pea protein on muscle damage because they did not mention about whey protein in their introduction. If that is correct, the authors should use whey protein with the same amount of leucine supplementation (8.2 g/100 g protein) as a positive control. If pea protein will have equal or higher effects, the pea protein supplementation will be beneficial as the authors hypothesized. However, they used pea protein and whey protein which had a different amount of leucine (12.7 g/100 g protein); thus, the readers will confuse the reason why the authors used pea protein (8.2 g/100 g protein) in this study.
If the authors would like to focus on both pea protein and whey protein which includes the same amount of total protein, they should revise their introduction in order to understand the rationale for using pea protein and whey protein in this study for the readers.
RESPONSE:
In the introduction, in response to your concerns, we adapted the last statements as follows:
We utilized a randomized, parallel group design, and emphasized high amounts of supplemental whey and pea protein split into three doses per day for several days post-exercise. The three daily acute pea and whey protein doses (0.3 g/k) differed in leucine content but both provided amounts within the recommended leucine intake range of 700-3,000 mg [13]. Thus, we hypothesized that large daily doses (0.9 g/kg) of pea and whey proteins compared to water spread throughout each day of the 5-day period would counter exercise-induced muscle damage, DOMS, and performance decrements.
These statements explain that we focused on a comparison of pea and whey protein (as found in commercial products) with the hypothesis that despite a difference in leucine content there was still a sufficient amount of leucine even in pea protein to reduce post-exercise damage. The last statements in the conclusion section provide the practical application, and we adapted these to address your concerns:
"This study focused on a comparison of pea and whey protein (as found in commercial products) with the hypothesis that despite a difference in leucine content there was still a sufficient amount of leucine even in pea protein to reduce post-exercise damage. This hypothesis was not fully supported by the data."
